# IoT-LLM: Enhancing Real-World IoT Task Reasoning with Large Language Models

## Abstract

Large Language Models (LLMs) have demonstrated remarkable capabilities across textual and visual domains but often generate outputs that violate physical laws, revealing a gap in their understanding of the physical world. Inspired by human cognition—where perception is fundamental to reasoning—we explore augmenting LLMs with enhanced perception abilities using Internet of Things (IoT) sensor data and pertinent knowledge for IoT task reasoning in the physical world. In this work, we systematically study LLMs' capability to address real-world IoT tasks by augmenting their perception and knowledge base, and then propose a unified framework, IoT-LLM, to enhance such capability. In IoT-LLM, we customize three steps for LLMs: preprocessing IoT data into formats amenable to LLMs, activating their commonsense knowledge through chain-of-thought prompting and specialized role definitions, and expanding their understanding via IoT-oriented retrieval-augmented generation based on in-context learning. To evaluate the performance, We design a new benchmark with five real-world IoT tasks with different data types and reasoning difficulties and provide the benchmarking results on six open-source and close-source LLMs. Experimental results demonstrate the limitations of existing LLMs with naive textual inputs that cannot perform these tasks effectively. We show that IoT-LLM significantly enhances the performance of IoT tasks reasoning of LLM, such as GPT-4, achieving an average improvement of 65% across various tasks against previous methods. The results also showcase LLMs' ability to comprehend IoT data and the physical law behind data by providing a reasoning process. Limitations of our work are claimed to inspire future research in this new era.

## 1 Introduction

Recent advancements in large generative models have showcased their exceptional performance and versatility in handling complex tasks across textual and visual domains, as evidenced by the GPT series (Radford et al., 2018; 2019; Brown et al., 2020; Achiam et al., 2023; OpenAI, 2023) and visual generation models (Dosovitskiy et al., 2020; Liu et al., 2021; Ho et al., 2020; Peebles & Xie, 2023; Blattmann et al., 2023). However, these models could occasionally generate outputs that are physically implausible, often referred to as "hallucinations" (Alkaissi & McFarlane, 2023; Huang et al., 2023). Even advanced video generation models, e.g., Sora (Brooks et al., 2024), are susceptible to producing animations that contravene fundamental physical laws, such as a video clip containing a tipping water glass that appears to defy gravity. These observations suggest that generative models may not really comprehend and apply physical laws of the physical world as accurately as humans when acting as world simulators. This has renewed interest in research on the *World Model* that focuses on understanding and modeling the physical world in a brain-like manner (Dawid & LeCun, 2023; Garrido et al., 2024; Mendonca et al., 2023; Liu et al., 2024).

Unlike Large Language Models (LLMs) that map descriptions of the physical world to a latent space and perform reasoning by predicting the text sequence according to the probability, research on human cognitive science illustrates a different mechanism. The human brain comprises multiple mutually-functional areas, of which the important components include the temporal and occipital lobes for perception, and the frontal cortex for reasoning (Churchland & Sejnowski, 1988; Saxe et al., 2009; Hobeika et al., 2016; Grèzes et al., 2001). Notably, perception is the primary mechanism through which information about the physical world is acquired, and then effective reasoning

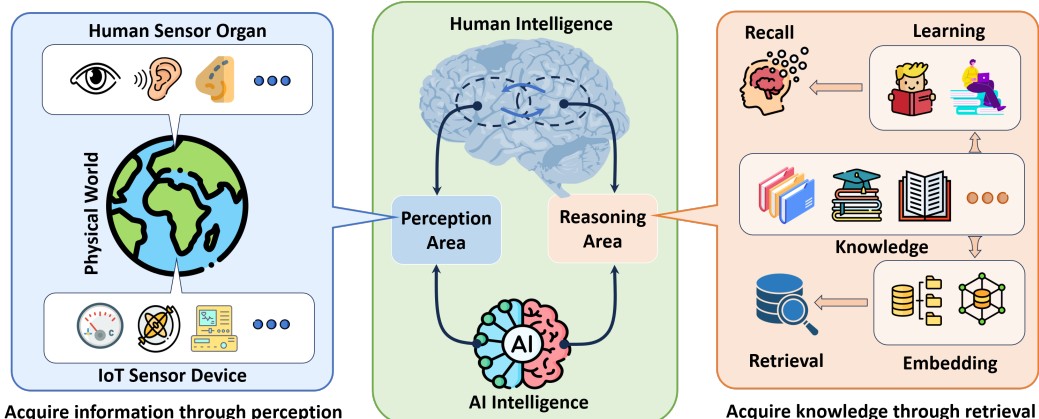

Figure 1: Inspired by human cognitive science, we augment LLMs with physical world perception from IoT data. Furthermore, by retrieving pertinent knowledge about IoT tasks, we enhance the reasoning capabilities of LLMs in executing real-world applications.

is inherently dependent on accurate perception. However, in LLMs, the physical world is only "perceived" through natural language, i.e., concepts and words in the semantic space, which denotes an indirect representation and abstraction of the physical world. A recent study in Nature shows language is primarily a tool for communication rather than thought (Fedorenko et al., 2024), so reasoning the physical-world problem with only language is limited. To enable LLMs with better reasoning capability in the real world, perception is highly demanded. Recent research on Vison Language Models (VLMs) builds the connection between visual perception and languages (Zhang et al., 2024a), yet the vision is only one of the various perceptual modalities. Many aspects of the physical world are still not perceived by existing LLMs.

We draw inspiration from how humans understand the physical world: perception to acquire information and reasoning with relevant domain knowledge. Firstly, humans perceive the world via a multitude of sensory organs, such as eyes for sight and ears for hearing. To empower machine perception, Internet of Things (IoT) sensors are developed. Since the first IoT sensor was designed for Coke machines to count the number of bottles in the 1980s (Madakam et al., 2015), IoT sensors become the "sensory organs" of machines, modeling the physical world for machine automation. Secondly, humans understand the world via the perception data with domain knowledge gained from experience and education. Similarly, LLMs can learn domain knowledge of both the physical world and sensors from the context to have stronger reasoning capabilities by in-context learning. In this manner, as shown in Fig. 1, we believe perception data with pertinent knowledge can enable LLMs to address complex problems with IoT-enabled perception in the real world. In this work, we aim to explore the following questions: (1) What types of real-world tasks can LLMs perform via the IoT perception of the physical world? (2) How can we enhance the LLM capability to deal with real-world tasks? (3) Do LLMs truly understand perception data and apply knowledge to realize real-world tasks?

Previous studies have primarily shown the viability of using LLMs for IoT task reasoning Xu et al. (2024b); Ji et al. (2024), but we find that these studies are not carefully scrutinized. (1) These studies only focus on specific tasks, such as R-peak identification and action recognition. The choices of tasks are not comprehensive, and thus they lack a benchmark to evaluate the performances of the methods. (2) They directly input raw IoT data into LLMs for reasoning, but LLMs are not good at dense numerical data and calculation (Zhou et al., 2024; Gruver et al., 2024). (3) They only evaluate their effectiveness on close-source LLMs, and lack a comprehensive study of benchmarking open-source LLMs with different parameter size.

To bridge this gap and answer the questions we proposed, we conduct an in-depth investigation of how to utilize LLMs to perform various tasks in the physical world using IoT data. Firstly, we explore whether LLMs can solve IoT classification and regression problems by setting a new bench-

mark with five classic IoT tasks with different data and levels of difficulties, including human activity recognition, industrial anomaly detection, heartbeat anomaly detection, WiFi-based human sensing, and indoor localization. The benchmark covers scenarios of daily life, industrial applications, and medical care, which will be detailed in the experiments. Secondly, we enhance LLMs' reasoning capabilities with IoT data through three novel steps and consolidate three steps into IoT-LLM, a unified framework for IoT task reasoning. It is composed of three steps tailored for IoT reasoning: designing an LLM-friendly data format, activating knowledge by chain-of-thought prompting, and automatic IoT-oriented Retrieval-Augmented Generation (RAG) based on LLMs' in-context learning capability. Thirdly, to determine whether LLMs truly understand and then solve the task, we have LLMs generate analytical processes and analyze the reasonableness of the analytics. The analysis generated by IoT-LLM indicates that LLMs can provide a reasonable process of solving simple tasks, but their efficacy diminishes in more specialized domains like heartbeat anomaly detection. This performance disparity is attributable to the complexity of data and limited domain-specific knowledge inherent in LLMs.

In summary, our contributions are as follows:

- We systematically study how Large Language Models (LLMs) can address real-world problems by perceiving the physical world via IoT sensor data.

- We propose a unified framework to address IoT-related real-world problems, which enhances the capability of LLMs through three steps: IoT data simplification and enrichment, IoT-oriented knowledge retrieval, and prompt configuration. To the best of our knowledge, this is the first unified framework for IoT-related tasks in the physical world.

- We establish the first benchmark for IoT task reasoning, including five real-world tasks with various types of IoT data. We benchmark both open-source and close-source LLMs with different parameter size. Empirical results show that our IoT-LLM significantly improves the performances of all base LLMs on IoT tasks.

## 2 RELATED WORK

**ML/DL methods in IoT tasks.** The Internet of Things (IoT) sensors gather diverse data from the real world, such as tri-axial acceleration, electrocardiogram readings, WiFi signals, and pressure (Sehrawat & Gill, 2019). These data have empowered various human sensing tasks, including Human Activity Recognition (HAR) (Lara & Labrador, 2013), health monitoring like heartbeat and respiration anomaly detection (Mousavi & Afghah, 2018; Aytekin et al., 2022), and industrial applications such as machine operational state monitoring (Kong et al., 2023). Currently, these IoT data are primarily processed using traditional machine learning techniques, such as Support Vector Machines (SVM) and K-Nearest Neighbors (KNN) Algorithm (Alam et al., 2016; Luo et al., 2021), or deep learning methods (Li et al., 2021; Njima et al., 2019). These approaches build black-box predictors for specific tasks, yet each predictor only supports one task, and the task cannot be addressed with reasoning analysis, which motivates us to explore LLM for IoT tasks.

**LLMs in IoT tasks.** Existing literature on Large Language Models (LLMs) in IoT mainly regards LLM as a user interface or as coordinators in smart machines (Li et al., 2023; Cui et al., 2023; Du et al., 2023). However, in these studies, LLMs function as intermediaries and do not directly interpret IoT data to perform real-world tasks. Recent studies, such as Penetrative AI (Xu et al., 2024a) and HarGPT (Ji et al., 2024), have begun integrating IoT data into LLMs for specific tasks, leveraging their inherent knowledge bases. Despite these advancements, the exploration of LLMs processing IoT data remains nascent. Penetrative AI converts IoT data into textual and numerical formats for basic tasks like R-peak identification in ECG data, heavily relying on manually crafted expert knowledge, which limits automation and scalability. Similarly, HarGPT processes raw IMU data to recognize human activities using a chain of thought technique but is restricted to this specific data type and task, not demonstrating the broader applicability of LLMs. While these studies provide initial insights into using LLMs in the IoT domain, they do not offer a comprehensive framework that fully exploits LLM capabilities or systematically explores the interaction between LLMs and the physical world through IoT devices, which is the primary focus of our work.

# 3 METHODOLOGY

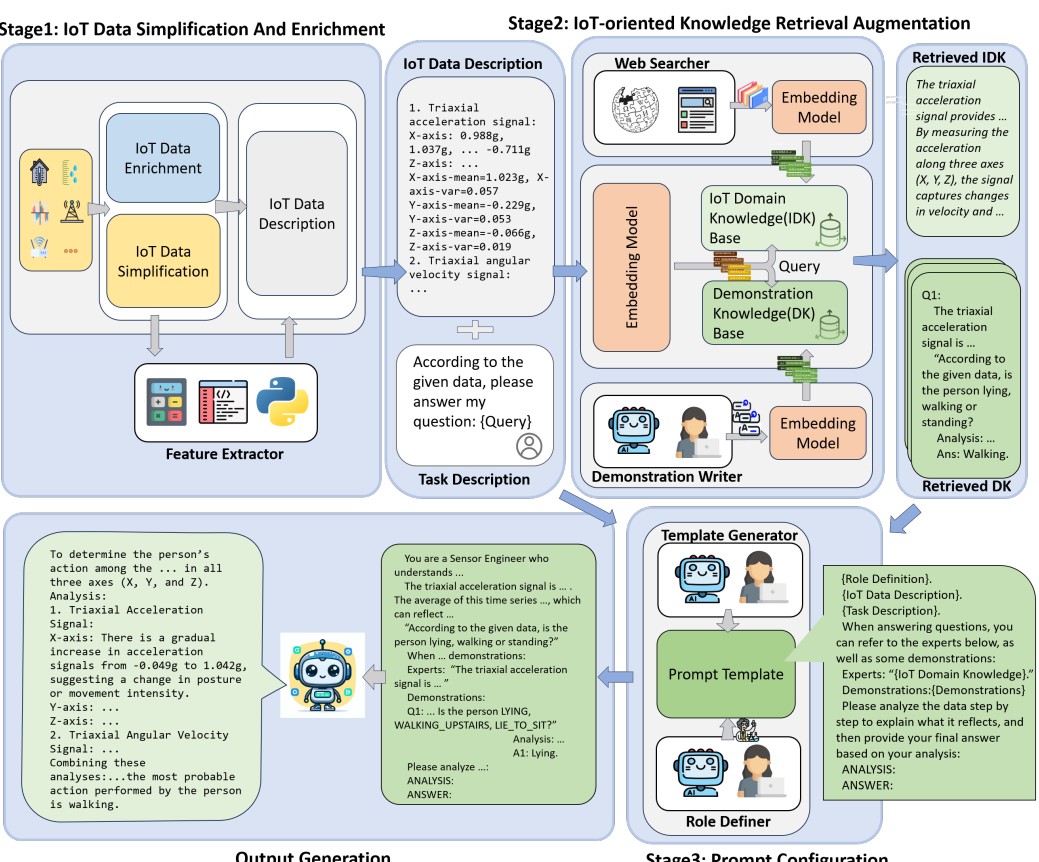

Figure 2: In our framework, IoT data is initially preprocessed to create a data description. Next, relevant IoT domain knowledge and task-specific demonstrations are retrieved. These elements are then combined into a prompt, which is input into a LLM to generate the final output.

In this section, we define the problem for IoT task reasoning with LLM and introduce our research methodology. The formulated research problem is how to leverage LLM and in-context learning for task reasoning for IoT data, termed as *IoT task reasoning*, e.g., using accelerators data for activity recognition or machine sensor for anomaly detection. The prompt for LLM should include two parts: data, as a way to perceive the physical world, and the task description, such as "Is it a Normal heartbeat (N) or Premature ventricular contraction beat (V)?", serves as the query. To evaluate the performance of IoT reasoning task, we build a new benchmark including 5 real-world tasks with different IoT data types and difficulty levels, encompassing both classification and regression problems.

At first, we employ LLMs to execute IoT tasks in a basic setting, similar to the existing approaches (Ji et al., 2024; Xu et al., 2024b), where the prompt provided to the LLMs includes only raw IoT data and the associated query. However, the performance of LLMs remains suboptimal. As shown by the baseline results in Table 2, even GPT-4 only achieves an accuracy of 43% for 3-way activity recognition and 50% for machine diagnosis based on their approach. These preliminary results akin to near-random guessing suggest a lack of comprehension of IoT data and tasks by this naive prompting way. Upon analyzing the characteristics of IoT data and real-world tasks, we identify that the challenges stem from the abstraction of dense numeric data and the lack of domain knowledge within LLMs. To address these challenges, we propose a unified framework (Fig. 2) consisting of three key stages: (1) IoT data simplification and enrichment, (2) IoT-oriented knowledge

augmentation, and (3) prompt configuration. Each stage addresses specific difficulties encountered by LLMs for IoT task reasoning, and we introduce each stage one by one.

### 3.1 IoT data simplification and enrichment

Unlike textual human tasks that have been learned by LLMs, IoT data for IoT task reasoning presents unique challenges that hinder LLMs' comprehension. Firstly, IoT data encompasses a diverse range of types and forms, many of which are complex time-series data (e.g., electrocardiogram readings) (Goldberger et al., 2000) or multi-variant data (e.g., WiFi CSI) (Yang et al., 2024). LLMs often struggle with accurately interpreting dense numerical data, especially when it involves long-sequence time-series data (Zhang et al., 2024b). Secondly, IoT data is typically composed of raw numerical values. This data often lacks essential textual annotations, such as units of measurement and metadata about the data collection process, which are critical for LLMs to interpret effectively in real-world applications. In summary, raw IoT data requires (1) appropriate simplification and (2) information enrichment. Previous studies (Xu et al., 2024b) have employed down-sampling techniques for time-series data but they only achieve coarse-grained simplification at a length level without enhancing the informational content of the IoT data. In contrast, we not only simplify IoT data at the token level but also enrich the IoT data by providing additional information to facilitate better understanding by LLMs (as illustrated in Fig. 10 in Appendix B). In this way, we transform complex raw IoT data into an LLM-friendly format for IoT task reasoning.

**IoT data simplification.** To achieve effective simplification, it is crucial to understand why LLMs struggle with dense numeric data. Firstly, according to recent research (Gruver et al., 2024; Spathis & Kawsar, 2023), tokenization methods, such as Byte Pair Encoding (BPE) often fragment numbers into tokens that do not align with their digits, resulting in inconsistent tokenization of floating-point numbers and complicating arithmetic operations. Therefore, in addition to down-sampling and keeping fixed precision (e.g., two digits of precision) to efficiently manage context length, we propose to insert spaces between digits to ensure distinct tokenization of each digit and use a comma (",") to separate each time step in a time series. Secondly, the complexity of long-sequence IoT data poses significant challenges for LLMs in analysis. To assist LLMs in processing this data, we extract essential statistical features, e.g., mean, variance, and FFT mean, utilizing external tools, such as Python scripts. We find that these fundamental features are strong enough for IoT task reasoning in classic IoT tasks. By doing so, we not only simplify IoT data at both length and token levels but also transform it into a format that is more suitable for tokenization and processing by LLMs.

**IoT data enrichment.** As previously noted, IoT data alone is insufficient for LLMs to effectively perform real-world tasks. To address this, we enrich the data by incorporating contextual information about the physical world. Specifically, we provide a comprehensive overview of IoT data collection and the integration of physical information. For instance, in human activity recognition (HAR) tasks where we employ inertial measurement unit (IMU) data including triaxial acceleration and angular velocity from accelerometers and gyroscopes, we meticulously outline the data collection process, incorporating the metadata such as sampling frequency (e.g., 10 Hz), device placement on the body, and units of measurement (e.g., gravitational acceleration and radians per second). This approach enables LLMs to not only align the three-axis IMU data with the corresponding three-dimensional spatial orientations of the human body but also to understand the physical significance of these numerical values, thereby enhancing the comprehension of LLMs for the task in the physical world.

### 3.2 IoT-oriented Knowledge Retrieval Augmentation

In IoT task reasoning, the knowledge of LLMs to perform IoT tasks is significant. For example, detecting abnormal heartbeats from electrocardiogram (ECG) data requires interpreting ECG signals and associating them with specific heartbeat states (e.g., normal, premature ventricular contraction), necessitating specialized domain knowledge. Although previous research (Xu et al., 2024b) proposes to include specific expert knowledge for specific tasks, the augmentation is task-specific and added manually, which is time-consuming and not scalable. To address this, we enable LLMs with IoT knowledge in an automatic fashion. Inspired by the in-context learning capability of LLMs, we also retrieve task-specific demonstrations, such as question-answer pairs, to guide LLMs in effectively utilizing IoT data for analyzing IoT tasks.

We first construct an IoT domain knowledge base and a demonstration knowledge base, which will be utilized for retrieving domain knowledge and task-specific demonstrations. To ensure comprehensive coverage of knowledge about IoT data and tasks within the IoT domain knowledge base, we gather relevant documents (e.g., Wikipedia articles, research papers) through web searches encompassing the following themes: (1) IoT data domain knowledge, (2) IoT task domain knowledge, and (3) expert insights on leveraging IoT data for task execution. For the demonstration knowledge base, we create task-specific demonstrations (i.e., question-answer pairs) authored by human or AI models (e.g., ChatGPT). We then employ an embedding model (e.g., text-embedding-ada-002[1] by OpenAI) to embed texts into vectors and store the text chunks and corresponding embeddings as key-value pairs, which allows for efficient and scalable search capabilities. To improve the quality of retrieved contents, we also store metadata (e.g., IoT data type for IoT domain knowledge base and task type for demonstration knowledge base) alongside the vector embeddings within the vector database. This approach allows for advanced post-processing techniques, such as metadata filtering (Poliakov & Shvai, 2024), to refine search results and improve task-specific retrieval accuracy. Secondly, we retrieve relevant knowledge using both IoT data description and task description as query. We adopt a hybrid search method, which means utilizing both keyword-based retrievers and embedding-based retrievers to harness their unique strengths, ensuring the consistent retrieval of highly relevant and context-rich information. Finally, after applying a re-ranking technique to recalibrate the similarity between the query and retrieved texts using ranker models (e.g. bge-reranker-base[2]), we filter out the top-m most relevant pieces, thus obtaining pertinent knowledge, encompassing documents with specific domain knowledge and task demonstrations relevant to the task at hand.

### 3.3 PROMPT CONFIGURATION

In addition to augmenting LLMs' knowledge by providing external documents in the context utilizing the in-context learning capability of LLMs, we further invoke LLMs' internal knowledge by carefully configuring the prompt. Recent studies demonstrate that LLMs possess strong role-playing capabilities (Park et al., 2023). To leverage this, we assign specific roles to LLMs for particular tasks. For instance, we have LLMs assume the role of a professional doctor when performing heartbeat anomaly detection, thereby activating their internal domain knowledge. What's more, since LLMs' reasoning capability can be improved a lot by decomposing the whole problem into several parts (Wei et al., 2022), we decompose the reasoning procedure into two steps, prompting LLMs to analyze the IoT data and task first, and then provide the final answer based on this analysis. By doing so, we can also evaluate the extent to which the LLM understands IoT data and its capability to perform IoT tasks through the generated analysis. In the end, we employ a prompt template (refer to Fig. 8 in Appendix B) to structure the content discussed previously. The ultimate prompt is crafted based on the template and subsequently fed into a downstream LLM. The LLM then produces the final output, encompassing both analysis and answer to the specified task.

## 4 EXPERIMENTS

### 4.1 A BENCHMARK ON IOT TASK REASONING

#### 4.1.1 IOT TASKS.

To comprehensively assess the capability boundaries of LLMs for IoT task reasoning, we develop a new benchmark comprising five real-world tasks with diverse IoT data types and difficulty levels: (1) Human Activity Recognition (HAR) using Inertial Measurement Unit (IMU) data, (2) Industrial anomaly detection using metrics such as temperature, cooling power, and cooling efficiency, (3) Heartbeat anomaly detection using Electrocardiogram (ECG) data, (4) Human sensing using WiFi Channel State Information (CSI), and (5) Indoor localization based on WiFi signal strength. It is important to note that we don't need to construct a knowledge base for each task especially, instead, we just need to construct two knowledge bases (i.e., one IoT domain knowledge base and one demonstration knowledge base), each of which contains all the domain/demonstration knowledge about the total five tasks. During the retrieval phase, we can easily fetch pertinent knowledge precisely corresponding to the task utilizing metadata (e.g., IoT data type and task type) stored within

---

[1] https://platform.openai.com/docs/guides/embeddings
[2] https://huggingface.co/BAAI/bge-reranker-base

the bases. For demonstrations, we utilize the one-shot setting, which means we retrieve one example for each category in classification tasks.

Table 1: **Performance of LLMs on WiFi-based Indoor Localization task.** Since this is a regression task, we choose the Root Mean Square Error (RMSE), Mean Absolute Error (MAE), and standard deviation (STD) of the RMSE as the main performance metrics.

| Method | | Model | | | | | |
|---|---|---|---|---|---|---|---|
| | | Llama2-7B | Mistral-7B | Claude-3.5 | Gemini-pro | GPT-3.5 | GPT-4 |
| Base-line | RMSE (m) | 0.374 | 11.570 | 0.829 | 2.318 | 2.598 | 0.741 |
| | MAE (m) | 0.313 | 9.347 | 0.696 | 1.814 | 1.937 | 0.581 |
| | STD | 0.903 | 6.856 | 1.607 | 5.999 | 6.715 | 1.502 |
| Ours | RMSE (m) | 0.355 | 9.995 | 0.404 | **0.313** | 0.719 | 0.402 |
| | MAE (m) | 0.295 | 7.980 | 0.341 | **0.265** | 0.592 | 0.341 |
| | STD | 0.852 | 11.146 | **0.706** | 0.763 | 1.765 | 0.697 |
| Impro-vement | RMSE (m) | **+5.1%** | **+13.6%** | **+51.3%** | **+86.5%** | **+72.3%** | **+45.7%** |
| | MAE (m) | **+5.8%** | **+14.6%** | **+51.0%** | **+85.4%** | **+69.4%** | **+41.3%** |

Table 2: **Overall performance of LLMs on IoT tasks**. **HAR-2cls** stands for classifying walking and standing activities. **HAR-3cls** stands for classifying lying, walking upstairs, and transitioning from lying to sitting activities. **Heartbeat** stands for classifying normal and abnormal heartbeats. **Machine** stands for determining whether the coolers work properly or not. **Occupancy** stands for detecting the presence of a person in a room.

| Model | | IoT tasks (Accuracy ↑) | | | | |
|---|---|---|---|---|---|---|
| | | HAR-2cls | HAR-3cls | Heartbeat | Machine | Occupancy |
| Llama2-7B | Baseline | 50.0% | 32.8% | 50.0% | 35.0% | 48.4% |
| | Ours | 57.2% | 38.0% | 54.5% | 56.4% | 82.5% |
| | Improvement | **+14.4%** | **+15.9%** | **+9.0%** | **+61.1%** | **+70.5%** |
| Mistral-7B | Baseline | 61.5% | 26.0% | 44.0% | 31.5% | 50.0% |
| | Ours | 84.9% | 42.7% | 60.5% | 92.1% | 61.1% |
| | Improvement | **+38.0%** | **+64.2%** | **+37.5%** | **+192.4%** | **+22.2%** |
| Claude-3.5 | Baseline | 98.3% | 80.1% | 52.4% | 51.0% | 50.0% |
| | Ours | **100.0%** | **95.3%** | **81.0%** | 86.3% | 82.5% |
| | Improvement | **+1.7%** | **+19.0%** | **+54.6%** | **+69.2%** | **+65.0%** |
| Gemini-pro | Baseline | 39.3% | 34.0% | 52.0% | 49.0% | 55.9% |
| | Ours | 88.4% | 82.8% | 51.5% | 70.1% | 66.2% |
| | Improvement | **+124.9%** | **+143.5%** | **-1.0%** | **+43.1%** | **+18.4%** |
| GPT-3.5 | Baseline | 91.5% | 33.3% | 35.3% | 51.5% | 50.0% |
| | Ours | 92.1% | 45.8% | 51.0% | 61.5% | **92.1%** |
| | Improvement | **+0.7%** | **+37.5%** | **+44.5%** | **+19.4%** | **+84.2%** |
| GPT-4 | Baseline | 77.3% | 43.3% | 54.0% | 49.5% | 43.7% |
| | Ours | **100.0%** | 87.8% | 69.8% | **92.4%** | 86.6% |
| | Improvement | **+29.4%** | **+102.8%** | **+29.3%** | **+86.7%** | **+98.2%** |

### 4.1.2 IoT DATASETS.

In our benchmark, we choose public IoT datasets on the five tasks to ensure fairness. Since some datasets are too challenging for LLMs with many classes, we simplify some datasets by only using a subset, which is also employed in previous works (Ji et al., 2024).

**Human Activity Recognition.**   We employ the Smartphone-Based Recognition of Human Activities and Postural Transitions Dataset (Reyes-Ortiz et al., 2015). This dataset comprises raw IMU data, specifically 3-axial linear acceleration, and 3-axial angular velocity, captured at a sampling rate of 50Hz by the smartphone's accelerometer and gyroscope. The data encompasses twelve distinct activities. To reduce both the sequence length and data complexity, we down-sample the data to 10Hz. Given the challenges associated with multi-class classification for LLM, instead of utilizing all twelve activity categories, we conduct a binary classification task involving the WALKING and STANDING labels, and a ternary classification task with the LYING, WALKING UPSTAIRS, and LIE TO SIT labels.

**Industrial anomaly detection.**   We employ the Condition Monitoring of Hydraulic Systems Dataset (Helwig et al., 2018), which facilitates the assessment of a hydraulic test rig's condition using multi-sensor data, including temperature, cooling power, and efficiency factor series, all experimentally derived from the rig. The dataset categorizes cooler conditions into three severity grades: (1) close to failure; (2) reduced efficiency; and (3) full efficiency. For simplicity, we focus on a binary classification task using only "close to failure"and "full efficiency"categories.

**Heartbeat anomaly detection.**   We employ the MIT-BIH Arrhythmia Database (Goldberger et al., 2000). This dataset comprises ECG recordings from 48 subjects, each sampled at 360Hz, and categorizes heartbeats into several types, including Normal beat (N), Atrial premature beat (A), and Premature ventricular contraction (V), among others. To reduce the difficulty of the task, we down-sample the signals to 72Hz and focus on a binary classification task using only the Normal beat (N) and Premature ventricular contraction (V) categories.

**Human sensing task.**   We utilize a dataset collected using a TP-Link TL-WDR4300 WiFi router operating at 5 GHz with a 40 MHz bandwidth (Zhuravchak et al., 2022). The dataset specifically captures the absence of human presence across three different rooms. Each room's environment is carefully monitored to record Channel State Information (CSI) that reflects the presence or absence of occupants, providing a robust basis for occupancy detection tasks.

**Indoor localization task.**   We utilize a dataset collected in a laboratory environment using an IoT system developed in  (Huang et al., 2022). The dataset consists of RSSI signals, the basis for determining human positions within the space. By collecting RSS fingerprints at various reference points, a signal radio map is constructed using a modified Gaussian Process Regression (GPR) method. This approach allows us to estimate the RSS distribution at any given location, providing a reliable means of localizing human presence in the environment.

### 4.1.3 LLM BASELINES.

In the conducted experiments, we utilize a combination of proprietary and open-source LLMs, including gpt-3.5-turbo, gpt-4-turbo, claude-3-5-sonnet, gemini-pro, Mistral-7B[3], and LLama2-7B[4]. This diverse selection of models enables a comprehensive evaluation of the LLMs' capabilities in executing IoT tasks and provides insights into their respective strengths and limitations in real-world applications. The code implementations of IoT-LLM have been attached in the supplementary materials and will be made public after publication.

### 4.2 RESULTS AND ANALYSIS

To evaluate the efficacy of our proposed framework in enhancing the capabilities of IoT task reasoning for LLMs, we use HarGPT (Ji et al., 2024) as the baseline, of which the prompts only contain

---

[3] https://huggingface.co/mistralai/Mistral-7B-Instruct-v0.3
[4] https://huggingface.co/togethercomputer/LLaMA-2-7B-32K

> 🔆 **Raw IoT sensor data:**
> 1. Triaxial acceleration signal:  X-axis: 0.189, 0.192,…, 0.194; Y-axis: 0.783, 0.786,…, 0.789;
> Z-axis: 0.582, 0.567,…, 0.562
> 2. Triaxial angular velocity signal:  X-axis: -0.013, 0.003, …-0.002;  Y-axis: -0.028, 0.004,…, 0.002;   Z-axis: -0.002, -0.004,…, -0.002

> 👤 **Query:**
> What activity is the subject most likely engaged in: lying down, walking upstairs, or lie-to-sit?

> 🤖 **Response from LLM:**
> **Analysis:**
> To analyze the given data and determine the most likely activity, I'll follow a step-by-step approach based on the expert knowledge and examples provided:
> **1. Acceleration Analysis:**
> - X-axis: The values start low (0.003g) and rapidly increase to around 1g. There's significant variation (var = 0.106) and a high mean (0.74g). This pattern suggests a change from horizontal to vertical orientation.
> - Y-axis: Values start high (around 0.9g) and decrease to negative values. High variation (var = 0.134) and changing direction indicate significant movement.
> - Z-axis: Values start positive (0.599g) and end negative (-0.21g). Moderate variation (var = 0.046) suggests some change in orientation.
> **2. Angular Velocity Analysis:**
> - X-axis: Fluctuating values with both positive and negative readings. Moderate variation (var = 0.185) indicates rotation around this axis.
> - Y-axis: Alternating positive and negative values with high variation (var = 0.286). This suggests significant rotational movement.
> - Z-axis: Highest variation among all axes (var = 0.437) with large positive and negative swings. Indicates substantial rotational movement around this axis.
> **3. Comparison with Examples:** The data pattern doesn't match the LAYING example, which showed very little variation. It's also different from WALKING_UPSTAIRS, which had more consistent, repetitive patterns. The data most closely resembles the LIE_TO_SIT example, with similar trends in acceleration and angular velocity changes.
> **4. Interpretation:** The data suggests a transition from a lying position to a sitting position: Initial low X-axis acceleration increasing to around 1g indicates a change from horizontal to more vertical orientation. Y-axis and Z-axis accelerations show changing orientations consistent with rising. High angular velocity variations, especially in the Z-axis, suggest the rotational movement involved in sitting up.
> **ANSWER: LIE_TO_SIT**
> **This activity best matches the observed data patterns, showing the characteristic transition from a lying to a sitting position with significant movement and rotational changes across all axes.**

Figure 3: Response example of LLM for human activity recognition (HAR). For more comprehensive examples, please refer to Appendix A.

raw IoT data and corresponding task descriptions, without any data preprocessing, domain knowledge, and demonstrations. The overall performance of LLMs on IoT tasks is shown in Table 1 and Table 2. The results show that our proposed method consistently boosts the performance of all the LLMs to complete IoT tasks in real-world scenarios. Notably, advanced LLMs such as Claude-3.5, Gemini-pro, and GPT-4 have demonstrated significant performance improvements, evolving from near-random guessing to effectively solving certain tasks. After analyzing the overall performance and outputs of LLMs in IoT task reasoning, we can answer the questions we proposed in the introduction now. Here is a summary of our arguments regarding the IoT task reasoning with LLMs.

**LLMs excel in various IoT tasks but struggle with complex data challenge.** Based on the experimental results, we observe that when provided with perception data (i.e., IoT data collected by sensors) and external knowledge, advanced LLMs like GPT-4 and Claude-3.5 can effectively perform various IoT tasks in the physical world, particularly excelling in HAR using IMU data. However, LLMs' performance is limited by their intrinsic lack of domain-specific knowledge and difficulty in comprehending numerical data. For instance, in the task of heartbeat anomaly detection, even provided with external knowledge, LLMs perform sub-optimally. This is because the time-series nature of ECG data presents significant challenges for LLMs due to its numerical density and length. Although we have mitigated some of these challenges by simplifying the data, this approach only addresses the issue at the data level without fundamentally resolving it at the model level. Additionally, LLMs inherently lack the extensive medical knowledge required for comprehensive analysis. While retrieved knowledge can suffice for simpler tasks, more complex problems may necessitate further model fine-tuning to incorporate deeper and broader medical expertise.

**LLMs are excellent learners in IoT task reasoning.** Without domain-specific knowledge and relevant demonstrations, LLMs face significant challenges in performing IoT tasks, often resorting to near-random guessing, especially in tasks such as heartbeat anomaly detection. This indicates that real-world tasks remain challenging for LLMs to execute directly. However, LLMs are excellent learners, and their capabilities can be significantly enhanced through data simplification & enrichment and knowledge retrieval augmentation. Specifically, the LLama2-7B, Mistral-7B, Claude-3.5, Gemini-pro, GPT-3.5, and GPT-4 models exhibit average performance improvements of 30%, 62%, 44%, 69%, 43%, and 65% respectively across various tasks, underscoring the effectiveness of our methodology.

**LLMs can act as experts, not just classifiers or predictors.** In our study, we prompt LLMs to generate both an analysis of the task and the final answer. Based on this analysis, we demonstrate that LLMs can fully comprehend preprocessed IoT data and effectively utilize the provided knowledge to perform IoT tasks. Unlike traditional DL/ML methods, which are trained end-to-end to produce only the final answer, LLMs offer more explainable results. Specifically, LLMs not only provide the final answer but also the reasoning behind it, akin to expert suggestions in daily life. For instance, when tasked with human activity recognition (as illustrated in Fig.3), the LLM delivers a detailed step-by-step analysis before presenting the final answer.

### 4.3 ABLATION STUDY

To evaluate the impact of different components within our framework, we performed an ablation study using GPT-4 on HAR and industrial anomaly detection tasks. We tested the following configurations: (1) IoT data simplification and enrichment, (2) addition of retrieved domain knowledge based on (1), (3) inclusion of retrieved demonstrations based on (2), and (4) the full configuration, which incorporates role descriptions and chain-of-thought techniques as outlined in the Prompt Configuration stage. The results, presented in Table 3, reveal that for straightforward tasks such as classifying walking and standing activities, IoT data simplification and enrichment and domain knowledge retrieval are sufficient. However, for more complex tasks, the inclusion of additional modules significantly boosts performance. Overall, our findings indicate that each module in our framework progressively enhances the ability of LLMs to perform IoT-related tasks using IoT data.

Table 3: Ablation study of different modules within our framework on three tasks.

| Method | IoT tasks (Accuracy↑) | | |
|---|---|---|---|
| | HAR-2cls | HAR-3cls | Machine |
| Baseline | 77.3% | 43.3% | 49.5% |
| *+ IoT data simplification and enrichment* | 96.0% | 47.3% | 62.7% |
| *+ retrieved domain knowledge* | 100.0% | 78.7% | 78.0% |
| *+ retrieved demonstrations* | 100.0% | 86.7% | 83.3% |
| Full setting | **100.0%** | **87.8%** | **92.4%** |

## 5 CONCLUSION

LLMs often struggle with tasks requiring an understanding of physical laws. To address this, we propose IoT-LLM, a framework that integrates IoT sensor data with LLMs to enhance their perception and reasoning abilities in the physical world. Evaluated on tasks like human activity recognition and industrial anomaly detection, IoT-LLM improves LLM performance by approximately 65%, though challenges remain in specialized domains. This approach systematically enhances LLM capabilities for real-world applications by leveraging IoT data.

**Limitations.** While LLMs can manage low-dimensional time-series data, they face significant challenges with higher-dimensional data, such as audio and 3D point cloud data, due to their extensive length and complexity. Integrating such data into the context of LLMs is both difficult and impractical. Instead, directly fine-tuning LLMs with data specific to these modalities may be a more effective approach. This strategy could be explored in future research to extend the capabilities of LLMs to handle more complex data types within the IoT domain.

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

# A    RESPONSE EXAMPLES OF LLMS FOR IOT TASKS

In this section, we present examples of responses generated by LLM for various applications, including industrial anomaly detection, heartbeat anomaly detection, WiFi-based human sensing, and indoor localization. To emphasize the output of the LLMs, we do not provide detailed prompts; instead, we simply display the raw IoT sensor data and the corresponding user queries.

---

🕐 **Raw IoT sensor data:**
1. Temperature Change Sequence: 57.277, … , 57.301
2. Cooling Power Change Sequence: 1.488, 1.497, …, 1.495
3. Cooling Efficiency Change Sequence: 18.704, … , 18.821

---

👤 **Query**

Is the machine's cooling system functioning properly?

---

🤖 **Response from LLM:**

To analyze the operational status of the machine's cooling system, we'll examine the Temperature Change Sequence, Cooling Power Change Sequence, and Cooling Efficiency Change Sequence step by step.

**1. Temperature Change Sequence Analysis:**
The sequence starts at 55.836°C and fluctuates throughout the 60-second period, with a slight increasing trend towards the end, closing at 55.75°C. The temperature shows a range of fluctuations but remains within a relatively narrow band (~55.5°C to ~56.0°C). This indicates that while there is a slight increase, the temperature does not significantly spike or drop, suggesting a potentially stable but slightly elevated operational temperature.

**2. Cooling Power Change Sequence Analysis:**
The cooling power starts at 1.568KW and shows variability throughout the sequence, peaking at 1.636KW and then decreasing towards the end, closing at 1.568KW. The cooling power's fluctuations are relatively minor, indicating that the cooling system is adjusting its output in response to the system's needs. However, the fact that it increases and then returns to the starting value suggests that the cooling system is actively working to manage the temperature, albeit with varying degrees of success.

**3. Cooling Efficiency Change Sequence Analysis:**
The cooling efficiency begins at 20.09% and exhibits some variability, peaking at 20.908% and then showing a decrease in efficiency towards the end, closing at 20.125%. The efficiency values indicate that the cooling system's effectiveness in removing heat from the system varies but reaches relatively high efficiency levels during the sequence. The decrease towards the end suggests a reduction in cooling performance over time.

**Final Analysis and Conclusion:**
The data reflects a cooling system that is actively working to manage the machine's temperature, as evidenced by the fluctuations in cooling power and efficiency. However, several indicators suggest that the cooling system may not be functioning optimally:
- The temperature, while relatively stable, is on the higher side throughout the sequence and shows a slight increasing trend. This suggests that the cooling system might be struggling to maintain or lower the temperature effectively.
- The cooling power shows that the system is adjusting its output, but the return to the initial value suggests that these adjustments may not be fully effective over time.
- The cooling efficiency, despite reaching relatively high levels, decreases towards the end of the sequence, indicating a potential drop in the cooling system's performance.
Based on this analysis, while the cooling system is operational and making adjustments in response to the machine's cooling needs, the slight increase in temperature, along with the variability in cooling power and the drop in efficiency towards the end, suggests that the cooling system may not be functioning at its best. It appears to be underperforming, possibly due to issues like partial blockages, reduced coolant flow, or other inefficiencies within the system.

**ANSWER: The machine's cooling system is not functioning properly and may require maintenance or further inspection to address potential issues affecting its performance.**

---

Figure 4: Response example of LLM for industrial anomaly detection.

💓 **Raw IoT sensor data:**

-0.205, -0.195, -0.155, -0.05, -0.015, 0.02, 0.125, 0.21, , ..., 0.055

👤 **Query:**

Is it a Normal heartbeat(N) or Premature ventricular contraction beat(V)?

🤖 **Response from LLM:**

To analyze the given ECG data, we will follow a systematic approach, examining the amplitude, intervals, and progression of the electrical signal that represents a heartbeat. This will involve looking for patterns that match either a normal heartbeat or a premature ventricular contraction (PVC).

**Step 1: Baseline and P-wave Identification**
- The initial segment shows a gradual increase in amplitude from -0.22mV to 0.125mV, which could represent the P-wave and the beginning of the QRS complex. In a normal heartbeat, the P-wave represents atrial depolarization. The gradual increase is consistent with the expected pattern, although the amplitude is somewhat low for a typical P-wave, which might not be clearly defined in this case.

**Step 2: QRS Complex Analysis**
- The amplitude sharply rises to 1.15mV, which is a characteristic feature of the QRS complex, representing ventricular depolarization. This sharp rise is expected in both normal and PVC beats. However, the amplitude of 1.15mV is within a normal range for the QRS complex, not indicating an overtly abnormal morphology based on amplitude alone.

**Step 3: Examination of the ST Segment and T-wave**
- After the peak, there's a dramatic drop to -1.395mV, which is highly unusual for a normal T-wave following the QRS complex. In a normal ECG, the T-wave represents ventricular repolarization and should be a positive deflection following the QRS complex, not a sharp negative drop. This significant negative deflection is not characteristic of a normal T-wave and indicates a significant abnormality.

**Step 4: Checking for Premature Beat**
- The dramatic changes in amplitude, especially the sharp drop after the peak, could suggest a PVC. PVCs are characterized by an abnormal QRS complex, which is wide and has an unusual shape. The drop to -1.395mV could be interpreted as part of an abnormally wide and bizarre QRS complex, which is a hallmark of PVCs. However, the presence of a preceding P-wave-like activity and the recovery back towards a more normal rhythm pattern complicate the interpretation slightly.

**Step 5: Final Analysis and Conclusion**
- The data shows a significant deviation from the normal ECG pattern, especially with the dramatic drop after the peak, which is not characteristic of a normal T-wave. The overall pattern suggests a significant abnormality in the ventricular repolarization process. Given the sharp rise indicative of the QRS complex followed by an abnormal repolarization phase, the pattern is more suggestive of a PVC due to the abnormal and dramatic changes in voltage that do not follow the typical progression of a normal heartbeat.

**ANSWER: Premature ventricular contraction beat (V)**
**This conclusion is based on the abnormal morphology of the waveform, particularly the unusual negative deflection following what appears to be the QRS complex, which does not align with the expected pattern of a normal T-wave. This analysis highlights the importance of examining the entire ECG waveform, including the amplitude, intervals, and progression of electrical activity, to differentiate between normal heartbeats and arrhythmias such as PVCs.**

Figure 5: Response example of LLM for heartbeat anomaly detection.

📶 **Raw IoT sensor data:**

[[0.12233546, 0.23833588, 0.21289147, 0.200783  , 0.35365783, 0.29699911, 0.19572163, …, 0.21170883],
 [0.11896763, 0.23593459, 0.21084769, 0.19843066, 0.31336457,0.29052874, … , 0.25554572, 0.24116531],
 …]

👤 **Query:**

Is there is a person or not?

🤖 **Response from LLM:**

To determine whether there is a person present based on the given data, we will analyze the provided values against the characteristics of the "no_person" and "have_person" classes.
**Given Data:**
1. **Mean value of CSI**: 0.282172876639514852.
2. **Standard deviation across subcarriers for the mean CSI amplitude over time**: 0.015816562357126943.
3. **Mean standard deviation across subcarriers for each time point**: 0.04446593535538511
**Reference Values**:
1. **No_person**:
Mean value of CSI: 0.29408252935437873 (larger than have_person)
Standard deviation across subcarriers for the mean CSI amplitude over time: 0.01709716445812625 (smaller than have_person)
Mean standard deviation across subcarriers for each time point: 0.03370765374037318 (smaller than have_person)
2. **Have_person**:
Mean value of CSI: 0.272173948247671 (smaller than no_person)
Standard deviation across subcarriers for the mean CSI amplitude over time: 0.022446105102973825 (larger than no_person)
Mean standard deviation across subcarriers for each time point: 0.051079445194106045 (larger than no_person)
**Step-by-Step Analysis**:
1. **Mean value of CSI**: The given mean value (0.28217287663951485) is between the mean values of "no_person" (0.29408252935437873) and "have_person" (0.272173948247671). This suggests a possibility of human presence, as it is closer to the "have_person" mean.
2. **Standard deviation across subcarriers for the mean CSI amplitude over time**:  The given standard deviation (0.01581656235712694) is smaller than the "no_person" standard deviation (0.01709716445812625) and also smaller than the "have_person" standard deviation (0.022446105102973825). This indicates that the variability is not increased, which is contrary to the expected behavior if a person were present.
3. **Mean standard deviation across subcarriers for each time point**:  The given mean standard deviation (0.04446593535538511) is larger than the "no_person" mean standard deviation (0.03370765374037318) but smaller than the "have_person" mean standard deviation (0.051079445194106045). This suggests some degree of variability, but it does not exceed the threshold expected for a "have_person" scenario.

**Conclusion**:
The analysis indicates that while the mean value of CSI is closer to the "have_person" category, the standard deviation across subcarriers for the mean CSI amplitude over time is lower than expected for a "have_person" scenario. Moreover, the mean standard deviation across subcarriers for each time point is not sufficiently high to classify it as "have_person."Given these observations, the data suggests that the environment is more likely consistent with the "no_person" category.
**ANSWER**: no_person

Figure 6: Response example of LLM for WiFi-based human sensing.

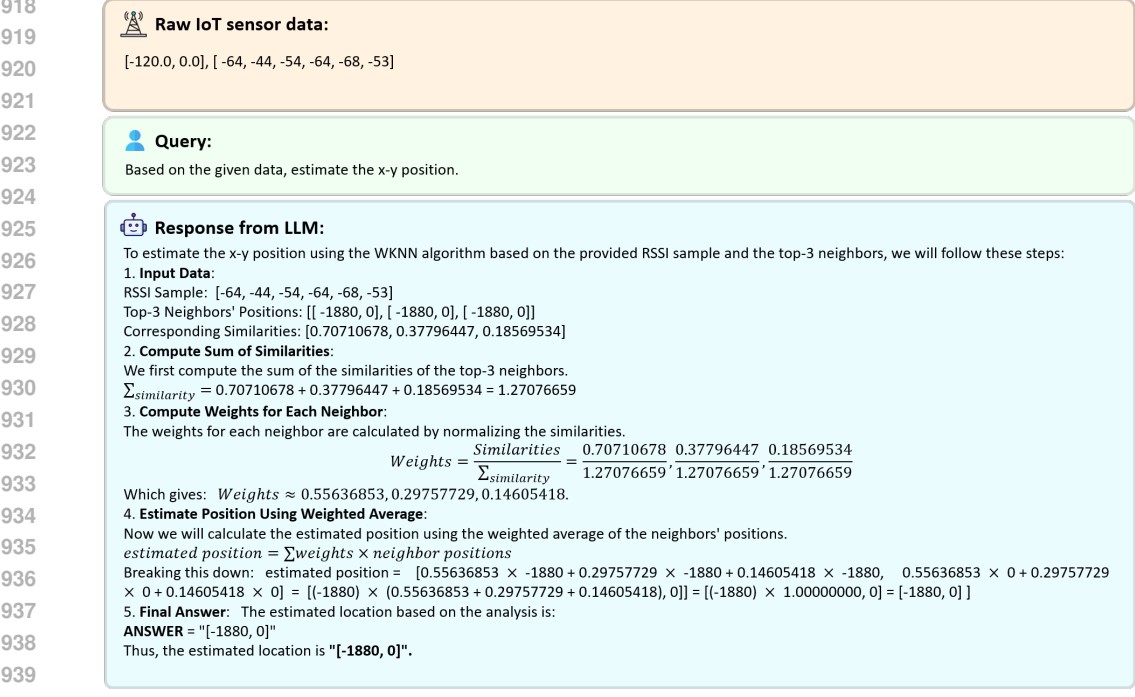

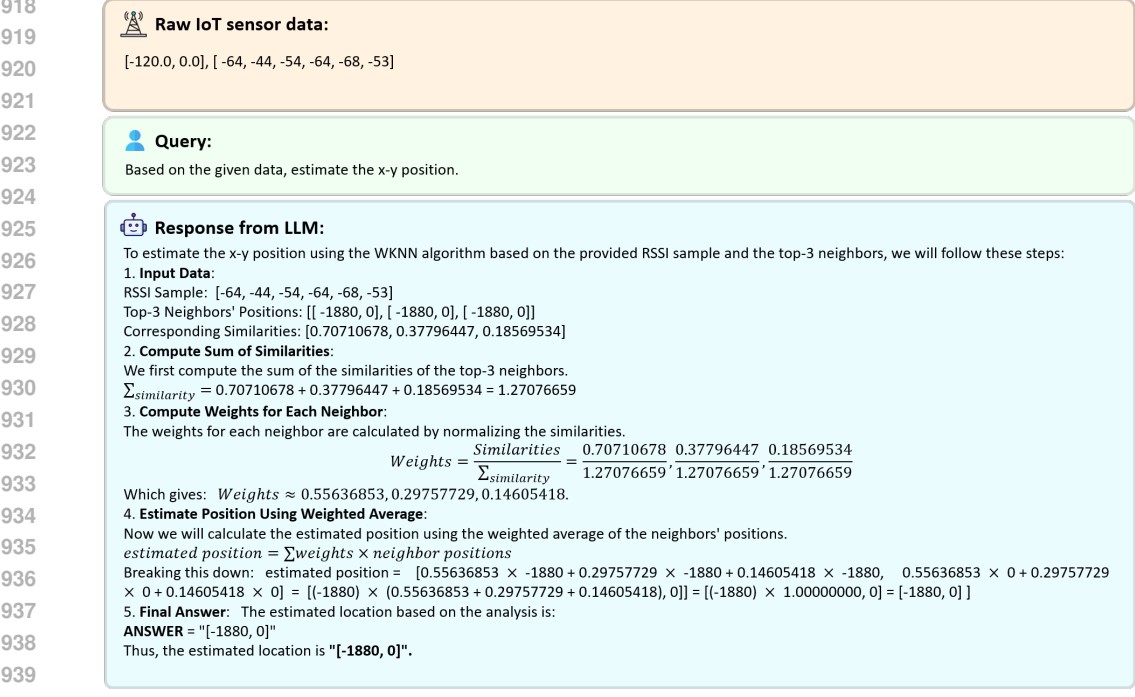

Figure 7: Response example of LLM for WiFi-based indoor localization.

## B  PROMPT TEMPLATE

In the Prompt Configuration stage within our framework, we systematically organize IoT data description, task description, retrieved pertinent knowledge (including IoT domain knowledge and task-specific demonstrations), and role description to generate the final prompt according to the prompt template, as shown in Fig. 8. For example, based on the final prompt template, we obtain the final prompt (as shown in Fig. 9) for heartbeat anomaly detection.

**{Role definition}**
The combined application of these knowledge and skills would enable you to accurately perform the task and provide relevant analysis and interpretation

EXPERT:
**{Retrieved domain knowledge}**

EXAMPLES:
**{Retrieved task-specific demonstrations}**

THE GIVEN DATA:
**{IoT data description}**

QUESTION:
**{Task description}**
Please analyze the data step by step to explain what it reflects, and then provide your answer based on your analysis.

ANALYSIS:
ANSWER:

Figure 8: Final prompt template.

You are an experienced physician who is familiar with various types of electrocardiogram (ECG) data. You can easily make preliminary judgments on whether heartbeats are abnormal based on ECG data. You possess the following medical and domain knowledge:

1. ECG Interpretation: You understand the basic principles of electrocardiography and know how to interpret ECG waveforms, including identifying different phases of the cardiac cycle and recognizing abnormalities.

2. Cardiac Physiology: You are familiar with the physiological functions of the heart, the generation and propagation of cardiac electrical signals, and the characteristics and manifestations of various cardiac arrhythmias.

3. Recognition of ECG Abnormalities: You are able to identify abnormal waveforms in ECG data, such as arrhythmias, myocardial ischemia, myocardial infarction, etc., and differentiate them from normal ECG patterns.

4. Medical Statistics: You are proficient in statistical analysis of ECG data, identification of outliers, and quantitative assessment of abnormalities.

5. Clinical Experience: You have extensive clinical experience to integrate ECG data with patient symptoms and medical history for accurate diagnosis and evaluation.

6. Medical Ethics and Legal Knowledge: You understand medical ethics and legal regulations to ensure confidentiality and lawful use of patient data.

The combined application of these domain knowledge and skills would enable you to accurately assess whether there are any abnormalities in the ECG data and provide relevant analysis and interpretation.

EXPERT:

Electrocardiography is the process of producing an electrocardiogram (ECG or EKG[a]), a recording of the heart's electrical activity through repeated cardiac cycles.[4] It is an electrogram of the heart which is a graph of voltage versus time of the electrical activity of the heart[5] using electrodes placed on the skin. In clinical applications, labeled ECG data are used to build a heartbeat classification system. Then this system is used to determine the types of heartbeats in unknown patients 'ECG recordings.

The overall magnitude of the heart's electrical potential is then measured from twelve different angles ("leads") and is recorded over a period of time (usually ten seconds). In this way, the overall magnitude and direction of the heart's electrical depolarization is captured at each moment throughout the cardiac cycle. A premature heart beat or extrasystole[1] is a heart rhythm disorder corresponding to a premature contraction of one of the chambers of the heart. Premature heart beats come in two different types: premature atrial contractions and premature ventricular contractions. Diagnosis Normal sinus rhythm and ectopic beats - premature ventricular contractions (PVC) and premature atrial contractions (PAC) shown on an EKG PVCs may be found incidentally on cardiac tests such as a 12-lead electrocardiogram (ECG/EKG) performed for another reason. In those with symptoms suggestive of premature ventricular complexes, the ECG/EKG is the first investigation that may identify PVCs as well as other cardiac rhythm issues that may cause similar symptoms. You can analyze whether the heartbeat is normal by considering a combination of factors such as the amplitude of peaks or valleys appearing in the electrocardiogram (ECG) time series, the time intervals between adjacent peaks or valleys, and the fluctuations in voltage values within the ECG data.

EXAMPLE1:

THE GIVEN ECG DATA:

-0.39mV, -0.38mV, -0.36mV, -0.355mV, -0.35mV, -0.37mV, -0.365mV, -0.35mV, -0.335mV, -0.35mV, -0.345mV, -0.355mV, -0.35mV, -0.33mV, -0.32mV, -0.295mV, -0.29mV, -0.295mV, -0.285mV, -0.23mV, -0.165mV, -0.08mV, -0.03mV, -0.1mV, -0.15mV, -0.185mV, -0.185mV, -0.16mV, -0.095mV, 0.325mV, 1.02mV, 0.53mV, -0.15mV, -0.22mV, -0.26mV, -0.305mV, -0.335mV, -0.31mV, -0.295mV, -0.275mV, -0.27mV, -0.245mV, -0.21mV, -0.145mV, -0.09mV, -0.03mV, 0.04mV, 0.105mV, 0.23mV, 0.365mV, 0.475mV, 0.52mV, 0.48mV, 0.375mV, 0.275mV, 0.16mV, 0.075mV, 0.0mV, -0.08mV, -0.135mV

ANSWER: Normal heartbeat (N)

EXAMPLE2:

THE GIVEN ECG DATA:

-0.55mV, -0.41mV, -0.29mV, -0.24mV, -0.16mV, -0.105mV, -0.08mV, -0.04mV, -0.055mV, -0.005mV, 0.085mV, 0.21mV, 0.42mV, 0.63mV, 0.785mV, 0.875mV, 0.9mV, 0.83mV, 0.705mV, 0.56mV, 0.405mV, 0.21mV, 0.125mV, 0.12mV, 0.08mV, 0.065mV, 0.06mV, 0.18mV, 0.48mV, 1.05mV, 1.57mV, 1.25mV, 0.81mV, 0.9mV, 0.05mV, -0.365mV, -0.525mV, -0.69mV, -0.76mV, -0.62mV, -0.79mV, -0.68mV, -0.685mV, -0.735mV, -0.785mV, -0.795mV, -0.82mV, -0.775mV, -0.7mV, -0.6mV, -0.485mV, -0.355mV, -0.24mV, -0.14mV, -0.115mV, -0.11mV, -0.115mV, -0.11mV, -0.085mV, -0.095mV

ANSWER: Premature ventricular contraction (V)

THE GIVEN DATA:

-0.205mV, -0.195mV, -0.155mV, -0.05mV, -0.015mV, 0.02mV, 0.125mV, 0.21mV, 0.345mV, 0.47mV, 0.615mV, 0.675mV, 0.705mV, 0.655mV, 0.6mV, 0.47mV, 0.36mV, 0.2mV, 0.135mV, 0.095mV, 0.045mV, 0.09mV, 0.025mV, 0.055mV, 0.05mV, 0.065mV, 0.14mV, 0.27mV, 0.29mV, 0.825mV, 1.35mV, 0.655mV, -1.18mV, -0.96mV, -0.89mV, -0.67mV, -0.47mV, -0.335mV, -0.27mV, -0.145mV, -0.12mV, -0.08mV, -0.1mV, -0.07mV, -0.005mV, 0.055mV, 0.155mV, 0.31mV, 0.52mV, 0.705mV, 0.875mV, 0.92mV, 0.865mV, 0.75mV, 0.6mV, 0.44mV, 0.245mV, 0.19mV, 0.125mV, 0.055mV

The ECG data is collected from a patient's heart. The data consists of a series of electrical signals that represent the heart's electrical activity. The signals are measured in millivolts (mV) and are recorded over a period of time at the sampling frequency of 60Hz. This means there is an interval of 0.017 seconds between the two voltage values. The data is divided into two categories: normal heartbeats (N) and ventricular ectopic beats (V). The normal heartbeats represent the regular electrical activity of the heart, while the ventricular ectopic beats represent abnormal electrical activity. The data is collected using a single-channel ECG device.

QUESTION:

Is the ECG heartbeat signal normal or abnormal?

Please analyze the data step by step to explain what it reflects, and then provide your answer based on your analysis.

ANALYSIS:
ANSWER:

Figure 9: Final prompt for heartbeat anomaly detection. Note that role description is generated automatically by AI models (e.g., ChatGPT).

1. (Triaxial acceleration signal) :
X-axis: 1.361 , 1.328 , 1.316 , 1.414 , 1.452 , 1.247 , 0.893 , 0.58 , 0.527 , 0.858 , 1.053 , 1.09 , 0.971 , 0.873 , 0.909 , 0.623 , 0.793 , 0.951 , 0.861 , 0.986 , 0.889 , 0.874 , 0.89 , 1.091 , 1.0 , 0.984 , 0.986 , 0.99 , 1.289 , 1.022 , 1.049 , 1.118 , 1.228 , 1.046 , 1.192 , 0.947 , 0.77 , 0.861 , 1.057 , 1.236 , 1.245 , 1.16 , 1.001 , 1.008 , 0.746 , 0.961 , 0.868 , 0.666 , 0.593 , 0.788 , 0.637 , 0.653 , 0.754 , 1.313 , 0.832 , 0.833 , 0.965 , 1.124 , 1.269 , 1.573 , 1.38 , 1.226 , 1.356 , 1.049 , 1.373 , 1.067 , 0.55 , 0.542 , 0.915 , 0.749 , 0.99 , 0.925 , 0.841 , 0.825 , 0.77 , 0.728 , 0.751 , 0.834 , 0.984 , 1.002 , 1.133 , 0.921 , 0.92 , 0.952 , 0.951 , 1.43 , 0.983 , 1.001 , 1.076 , 1.137 , 1.036 , 1.258 , 1.184 , 1.07 , 0.88 , 0.789 , 0.798 , 1.289 , 1.396 , 1.249 , 0.788 , 0.932 , 0.893 , 0.909 , 0.844 , 1.013 , 0.758 , 0.742 , 0.687 , 0.678 , 1.425 , 0.773 , 0.821 , 0.855 , 0.952 , 0.881 , 1.236 , 1.336 , 1.336 , 1.342 , 0.916 , 1.472 , 1.421 , 1.108 , 0.63 , 0.529 , 0.837 , 0.826
Y-axis: …
Z-axis:…
2. (Triaxial angular velocity signal) :
X-axis: -0.015, -0.317, -0.377, -0.268, 0.15, 0.883, 0.61, 0.07, -0.403, -0.8, -0.923, -0.661, -0.013, 0.323, 0.47, 0.446, 0.133, 0.238, 0.214, 0.262, 0.044, 0.017, 0.031, 0.072, 0.157, 0.588, 0.476, 0.509, 0.456, 0.292, 0.036, -0.009, 0.056, -0.075, -0.665, 0.39, -1.524, -0.833, 0.059, 0.743, -0.033, 0.305, -0.154, -0.562, -0.673, -0.374, -0.397, -0.352, -0.151, -0.082, -0.385, 0.199, 0.141, 0.261, 0.147, -0.182, -0.111, -0.378, -0.408, -0.401, 0.302, -0.505, -0.434, -0.018, 0.58, -0.717, 0.534, 0.145, -0.469, -1.032, 0.181, -0.559, -0.105, 0.37, 0.416, 0.498, 0.344, 0.424, 0.404, 0.261, 0.16, 0.124, 0.087, 0.21, 0.318, 0.371, 0.353, -0.957, 0.078, 0.101, 0.187, 0.113, 0.909, -1.101, -1.475, -1.093, -0.524, 0.16, -0.582, 0.467, 0.145, -0.085, -0.5, -0.243, -0.659, -0.794, -0.549, -0.435, 0.161, 0.009, 0.067, 0.18, 0.071, -0.084, -0.151, -0.0, -0.109, -0.049, -0.13, -0.289, -0.443, -0.452, 0.593, 0.798, -0.089, 0.074, -0.52, 0.922
Y-axis: …
Z-axis: …

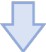

1. Triaxial acceleration signal:
X-axis: 1.3 6 1 g, 1.4 5 2 g, 0.8 5 8 g, 0.9 0 9 g, 0.8 6 1 g, 1.0 9 1 g, 1.2 8 9 g, 1.0 4 6 g, 1.0 5 7 g, 0.7 4 6 g, 0.7 8 8 g, 1.3 1 3 g, 1.2 9 6 g, 1.0 4 9 g, 0.7 4 9 g, 0.7 7 g, 1.1 3 3 g, 1.4 3 g , 1.0 3 6 g, 0.7 9 8 g, 0.7 8 8 g, 1.0 1 3 g, 1.4 2 5 g, 0.8 8 1 g, 0.9 1 6 g, 0.8 2 6 g
Y-axis: …
Z-axis: …
X-axis-mean=1.0 3 4 g, X-axis-var=0.0 5 3
Y-axis-mean=-0.1 6 9 g, Y-axis-var=0.0 3 2
Z-axis-mean=-0.0 0 7 g, Z-axis-var=0.0 4 7
2. Triaxial angular velocity signal:
X-axis: -0.0 1 5 rad/s, 0.8 8 3 rad/s, -0.6 6 1 rad/s, 0.1 3 3 rad/s, 0.2 6 2 rad/s, 0.5 8 8 rad/s, 0.0 3 6 rad/s, 0.3 9 rad/s, -0.0 3 3 rad/s, -0.3 7 4 rad/s, -0.3 8 5 rad/s, -0.1 8 2 rad/s, 0.3 0 2 rad/s, -0.7 1 7 rad/s, 0.1 8 1 rad/s, 0.3 4 4 rad/s, 0.1 2 4 rad/s, -0.9 5 7 rad/s, 0.9 0 9 rad/s, -0.5 8 2 rad/s, -0.2 4 3 rad/s, 0.1 6 1 rad/s, -0.0 rad/s, -0.4 5 2 rad/s, -0.0 8 9 rad/s, 0.9 2 2 rad/s
Y-axis: …
Z-axis: …
X-axis-mean=0.0 2 1 rad/s, X-axis-var=0.2 3 5
Y-axis-mean=0.1 3 8 rad/s, Y-axis-var=0.5 0 7
Z-axis-mean=-0.0 26 rad/s, Z-axis-var=0.0 6 5

1. Triaxial acceleration signal:
The provided three-axis acceleration signals contain acceleration data for the X-axis, Y-axis, and Z-axis respectively. Each axis's data is a time-series signal consisting of some data samples, measured at a fixed time interval with a frequency of 10Hz(10 samples is collected per second) by smartphone (Samsung Galaxy S II) wearing on the waist. The unit is gravitational acceleration (g), equivalent to 9.8m/s^2. It's important to note that the measured acceleration is influenced by gravity, meaning the acceleration measurement along a certain axis will be affected by the vertical downward force of gravity.
2. Triaxial angular velocity signal:
The provided three-axis angular velocity signals contain angular velocity data for the X-axis, Y-axis, and Z-axis respectively. Each axis's data is a time-series signal consisting of some data samples, measured at a fixed time interval with a frequency of 10Hz. The unit is radians per second (rad/s).

Figure 10: During IoT data simplification and enrichment stage, raw IoT data is transformed into IoT data description, which is easier to understand by LLMs. Raw IoT data is enriched with descriptive metadata, including natural language expressions of implicit physical information like units. Specialized tokenization techniques and extraction of temporal or frequency domain features further enhance LLMs' understanding of numerical and time-series data. These improvements make IoT data more accessible and interpretable for LLMs, facilitating its use in real-world applications.

