# OpenReview forum: "IoT-LLM: Enhancing Real-World IoT Task Reasoning with Large Language Models"
_ICLR.cc/2025/Conference — ICLR 2025 Conference Withdrawn Submission_

### Official Review · Reviewer_DDEh · 2024-10-27

**Soundness:** 2
**Presentation:** 2
**Contribution:** 2
**Rating:** 3
**Confidence:** 4

**Summary:**

The paper introduces a framework that leverages IoT sensor data to improve LLMs’ ability to handle physical-world tasks. It tackles challenges through data simplification, data enrichment, and knowledge retrieval. Benchmark results on various sensory tasks, such as activity recognition and anomaly detection, show up to 65% performance improvement. The paper highlights limitations with high-dimensional data, suggesting future work on model fine-tuning.

**Strengths:**

1. This paper addresses a timely topic

2. It expands the real-world applications of LLMs

3. This paper provides a thorough evaluation of five real-world IoT tasks with six LLMs.

4. Consistently improved accuracy.

**Weaknesses:**

1. Most importantly, the technical components proposed in this paper lack novelty when compared to existing studies. For example, state-of-the-art approaches for multimodal LLMs (MLLMs) incorporate visual inputs, which outperform the text-only prompts used in this work [r1, r2].

2. Additionally, the concept of “IoT data simplification” is adopted from prior research (Gruver et al., 2024; Spathis & Kawsar, 2023), and feature extraction is also widely utilized in existing studies. The use of “IoT data enrichment” and “knowledge retrieval” similarly aligns with a recent study [r1]. Given that text-only prompts are no longer the most effective approach, and many of the proposed ideas are derived from prior work, I do not see substantial technical novelty in this study.

3. As a suggestion, the term “IoT task” feels too broad and potentially misleading. This work primarily focuses on sensory data, so it may be more appropriate to refer to these as “sensory tasks.” The term “IoT tasks” suggests broader applications, such as controlling IoT devices via networking.


References:
[r1] By My Eyes: Grounding Multimodal Large Language Models with Sensor Data via Visual Prompting (EMNLP’24)
[r2] Plots Unlock Time-Series Understanding in Multimodal Models (arXiv ’24)

**Questions:**

Please see the weaknesses.

---

### Official Review · Reviewer_nZFU · 2024-10-28

**Soundness:** 2
**Presentation:** 3
**Contribution:** 2
**Rating:** 3
**Confidence:** 4

**Summary:**

This paper proposes a IoT-LLM, to augment LLMs for better processing and reasoning with IoT sensor data for real-world applications. It introduced a three-stage to enhance the LLM capability in handling IoT scenarios: (1) simplification and enrichment of IoT data, (2) chain-of-thought prompting and (3) RAG with in-context learning. It also develops a benchmark comprising five diverse IoT tasks  and evaluate the performance of several open- and closed-source LLMs. Results indicate improvements in IoT task accuracy across different models.

**Strengths:**

1. The paper offers a fresh approach by integrating IoT data directly into LLMs, which traditionally rely on textual data, to enhance real-world task reasoning.
2. This paper test the solution with various IoT tasks, proving its effectiveness.

**Weaknesses:**

1. The paper should address a fundamental question: why should we use LLMs for IoT data tasks, which are typically numerical in nature? While LLMs excel at handling textual data, they are not known for expertise in processing numerical data. Given the high cost associated with LLMs, what unique advantages do they offer for IoT tasks, especially when many other solutions are already available?

2. The paper lacks a baseline comparison with traditional approaches for IoT tasks. Without evaluating the gains or advantages of LLMs over these conventional methods (e.g., in terms of accuracy, efficiency, or cost), the benefits of using LLMs for IoT remain unclear.

3. Although the paper incorporates certain adaptations, such as Chain-of-Thought (CoT) and Retrieval-Augmented Generation (RAG) to make LLMs work for IoT tasks, these methods are standard. The paper’s novelty appears limited in this regard, and it would benefit from additional details and examples to illustrate precisely how these steps are implemented.

**Questions:**

1. What is the unique advantages of using LLMs to address the IoT tasks?
2. How does LLMs compare with non-LLM approaches?
3. Can the authors provide more details or examples on how the data processing, CoT and RAG work? Does these steps generalize or need manual design for different tasks?

---

### Official Review · Reviewer_3HQs · 2024-11-01

**Soundness:** 2
**Presentation:** 3
**Contribution:** 1
**Rating:** 3
**Confidence:** 4

**Summary:**

This work proposes new methods for using LLMs to perform real-world tasks using IoT data. They suggest various new benchmarks for the evaluation of LLMs and outline a method for improving the reasoning capabilities of an LLM for IoT task reasoning.
This work is a good start and studies an interesting topic, but I feel the experimental setting and analysis is lacking. More information is needed about the methodology, and a deeper analysis of the results is required. There should also be a discussion tackling the practicality of using an LLM in an IoT setting, where high-frequency predictions are required. These results should also then be compared to the classic ML methods discussed in the related work.

**Strengths:**

- Well written paper that discusses the problem and current solutions well.
- Clearly presented framework for using LLMs with IoT data.
- Improving the capabilities of language models in IoT is an interesting problem as it serves to increase their adoption. It also might improve the uptake of language models in IoT scenarios where LLM responses can be probed with natural language.
- A small ablation study is performed.

**Weaknesses:**

- The experimental results are limited. The only experiments in the paper are 5 classification tasks and 1 regression task using 5 LLMs. This does not provide much variety in the experimental setting and the analysis can be expanded upon. For example:
* Does the language model consistently provide reasoning for its predictions that align with the correct reasoning?
* Why does the language model perform the worst on Heartbeat?
* How does the amount of relevant literature on the IoT task affect the results? For example, what happens if we had very little literature to retrieve for the Occupancy task?
* How much of each of the datasets has the language model memorised?
* What happens if there is class imbalance? Is there class imbalance in the datasets and if so is accuracy the correct metric?
* How is the performance impacted by the number of demonstrations?
* Why does Gemini-Pro show a decrease in performance on Heartbeat?
* How do the results compare to the classical models currently used as best practise in these settings?

- There is a lot of missing experimental information, such as:
* How many repeats were done for the regression task?
* Were any repeats done for the classification tasks? Since no STD is provided.

- There should also be a discussion about the practicality of using an for IoT tasks, where compute is likely limited and costs of API calls could be significant if the model needs to make predictions continuously at a high frequency. For example:
* What is the cost of using GPT-4 or 3.5 for the heartbeat dataset (which is sampled at 72Hz)?
* Is it possible to use an LLM to make predictions on this dataset at 72Hz using an API?
* How much compute is required to use one of the open source models the authors present at 72Hz?

**Questions:**

1. Please discuss the practicality and cost of using an LLM in an IoT setting, where high frequency predictions are required.
2. Please compare the results with the classic ML methods that are mentioned in the related work.
3. Consider the performance of this method in settings with access to few demonstrations (this might be where IoT-LLM beats classic ML!), or with little domain knowledge (maybe where classic ML wins?).

---

### Official Review · Reviewer_pG4X · 2024-11-02

**Soundness:** 2
**Presentation:** 3
**Contribution:** 3
**Rating:** 5
**Confidence:** 5

**Summary:**

This paper presents a framework that leverages Large Language Models (LLMs) to tackle a set of representative IoT tasks: Human Activity Recognition (HAR), industrial anomaly detection, heartbeat anomaly detection, human sensing through WiFi Channel State Information (CSI), and WiFi-based indoor localization. The authors aim to demonstrate the framework’s generalizability by applying it to these varied applications, each characterized by different sensor data types and operational requirements. Key components of the framework include steps for data enrichment and domain-specific knowledge retrieval, which are designed to improve interpretability and predictive accuracy across tasks. The ablation study shows how each module contributes to performance gains, providing support for the framework’s design. By evaluating the framework across multiple LLMs and using metrics appropriate to each task, such as RMSE and MAE for regression, the authors aim to establish its versatility and effectiveness in real-time, sensor-driven IoT contexts.

**Strengths:**

- **Originality:** This paper contributes a structured framework for applying LLMs to IoT tasks across several real-world applications, including Human Activity Recognition (HAR), industrial anomaly detection, heartbeat anomaly detection, human sensing using WiFi CSI, and WiFi-based indoor localization. While the general approach of using LLMs in IoT contexts isn’t groundbreaking, the framework itself is a notable addition. Its modular design is carefully tailored for IoT applications, with components specifically aimed at data enrichment and domain-specific knowledge retrieval. The originality here lies less in the overall concept and more in how the framework is configured, particularly the focus on effective prompt structures. The work adds value by addressing prompt design—what structures are appropriate, how to populate templates, and why this approach is suitable for IoT applications. The authors convincingly show that their prompt template generalizes across IoT tasks within a certain set of data properties and task types.

- **Quality:** The framework is well-constructed, and the ablation study demonstrates the individual contributions of each module. However, the quality is compromised by a few overclaims and issues with baseline selection. For instance, the Penetrative AI prompt provides a more structured and informative context for LLM-based IoT reasoning, which the authors do not adequately address in their comparison. This choice weakens the evaluation, as it doesn’t fully account for how the proposed prompt structures stack up against more sophisticated baselines. Despite this, the modular evaluation and rationale behind each component provide useful insights into designing IoT-specific frameworks for LLMs, making it a worthwhile contribution.

- **Clarity:** The paper is very well-written, with a clear structure and accessible presentation of complex concepts. Each design choice is well-motivated, and the breakdown of each module’s role within the framework makes the overall structure easy to follow. The presentation of results, including metrics like RMSE and MAE for tasks that require them, is done with precision and supports the authors’ claims effectively. This clarity is a strength, helping readers understand not only the framework’s structure but also the logic behind each component.

- **Significance:** This paper makes a modest but important step toward making LLMs relevant for IoT data applications. While it’s not a transformative leap, the framework provides a modular approach that future work can build on, particularly in designing IoT-specific prompt structures. The work underscores the importance of prompt design in enhancing the applicability of LLMs to IoT tasks, particularly in real-time, sensor-driven contexts. That said, the paper’s impact is tempered by a lack of precision in articulating the problem space and explicitly linking IoT data properties to the modular design choices. Still, this is an incremental contribution in the right direction for LLM-based IoT frameworks.

**Weaknesses:**

1. **The baseline comparison over-represents the margin of improvement** by setting up a simplified baseline that doesn’t accurately reflect the improvement over the previous best framework, particularly the Penetrative AI prompt. The current baseline acts as a “strawman,” making the gains seem larger than they might be if compared against a more structured baseline. The Penetrative AI prompt, for instance, already includes a rich, context-aware structure and could likely perform even better if it incorporated the information retrieval elements proposed in this IoT LLM paper. To present a more balanced view of the framework’s improvements, the authors should consider comparing against a baseline like Penetrative AI, augmented with similar components, to better showcase the incremental gains of the new approach.

Here's the Penetrative AI prompt structure:
--
Objective:
[Clear task statement]

Background Knowledge:
[Expert domain knowledge]
[How to interpret different data types]

Response Format:
Reasoning: [Analysis instructions]
Summary: [Summary format]
[Specific output requirements]

[Optional: Processing Procedure]
[Step-by-step analysis instructions]

[Optional: Reasoning Example]
[Sample data]
[Example reasoning]
[Example outputs]

Query Data:
[Actual sensor data for analysis]
--

This is the HarGPT Prompt structure:
--
### Instruction:
You are an expert of IMU-based human activity analysis.

### Question:
The IMU data is collected from {device name} attached to the user's {location} with a sampling rate of {freq}. The IMU data is given in the IMU coordinate frame. The three-axis accelerations and gyroscopes are given below.

Accelerations:
x-axis: {...}, y-axis: {...}, z-axis: {...}

Gyroscopes:
x-axis: {...}, y-axis: {...}, z-axis: {...}

The person's action belongs to one of the following categories: <category list>.

Could you please tell me what action the person was doing based on the given information and IMU readings? Please make an analysis step by step.

### Response: {answer}
--

These were extracted from each respective paper.

2. **The paper tends to over-claim the novelty and impact of its approach.** While the modular design and tailored prompt structure are thoughtful, the idea of applying LLMs to IoT tasks isn’t entirely new—similar efforts, such as Penetrative AI and HardGPT, have covered comparable ground. Reframing this work as an incremental improvement, rather than a fundamentally new approach, would provide a more accurate context. Clarifying the unique aspects of this framework, particularly in terms of modular design for IoT tasks, would help in accurately positioning its contributions.

3. **The problem formulation lacks precision, and the specific properties of IoT data that drive the modular design choices are not clearly articulated.** Although the framework includes components like data enrichment and domain knowledge retrieval, the connection between these components and IoT-specific data characteristics (such as sensor noise, temporal dependencies, or data sparsity) is not fully articulated. A clearer problem formulation that explicitly links these IoT-specific data properties to each module would strengthen the motivation behind the design choices and make the framework’s modularity more compelling.

4. **The authors claim that the prompt structures generalize across various IoT tasks, but the evaluation doesn’t fully substantiate this.** The ablation study shows some adaptability, but it’s limited in scope and doesn’t fully demonstrate generalizability across a broader range of data types and tasks. To their credit, they do articulate this in the limitations, but expanding the evaluation to include additional tasks or providing further evidence of prompt adaptability would make their generalization claims more sound.

5. **The paper’s main contribution centers on IoT-specific prompt design, yet the reasoning behind each prompt structure isn’t deeply examined.** Although the prompts are functional, the paper lacks a clear explanation for why certain components are emphasized or why specific details are included. It would strengthen the work if the authors provided a more detailed analysis of each prompt element’s necessity, how it aligns with IoT task requirements, and how changes to the prompt structure might impact performance. This would clarify the contribution and serve as a useful guide for adapting the framework to other IoT scenarios. Basically, it is not clear how the properties of the data are directly linked to the prompt.

6. **The ablation study is useful but limited, as it covers only three out of the five tasks, which restricts its scope in evaluating the framework’s modularity.** A more complete ablation study across all tasks would provide a clearer view of each module’s contribution, potentially highlighting specific strengths and limitations in greater detail.

**Questions:**

1. **Why was the Penetrative AI prompt not used as a baseline or comparison?** Penetrative AI has a well-structured, context-rich prompt that could serve as an appropriate baseline for assessing the improvements in your framework. Additionally, it would be interesting to see how Penetrative AI would perform if it incorporated some of the information retrieval elements proposed in your framework, as it shares similar modular components. Could you clarify why Penetrative AI wasn’t included in the evaluation, and whether you considered augmenting it with your proposed elements to provide a more balanced comparison?

2. **How are specific properties of IoT data driving your modular design choices?** The framework includes various modules, such as data enrichment and domain knowledge retrieval, but it’s not entirely clear how these modules are directly motivated by the unique characteristics of IoT data (e.g., sensor noise, temporal dependencies, data sparsity). Could you explain how these IoT-specific data properties informed the inclusion and structure of each module in the framework? This connection would strengthen the rationale behind the modular design.

3. **How do the properties of IoT tasks drive the prompt structure, and how does the prompt address both shared and unique properties across tasks?** The paper highlights how the prompt structures generalize across various IoT tasks, but it doesn’t fully examine the specific properties—both implicit and explicit—that these tasks share or differ in, and how these properties guide the design of the prompts. For example, a breakdown of characteristics like multimodality, the spatial relationship of the sensor to the phenomena being measured, and temporal dependencies would clarify why certain prompt components are necessary. A table outlining these shared and distinct properties of IoT tasks, along with explanations of how the prompt structures address each, would strengthen the paper and justify the framework’s modularity. This would make it clearer how the prompts are designed to handle both the commonalities and the unique demands of each IoT task.

4. **How does the generalization claim hold across a wider range of IoT tasks?** The paper asserts that the prompt structures generalize well across IoT tasks, but the current evaluation covers only a subset. Is there additional evidence to support this generalization, or would you consider expanding the evaluation to a broader set of IoT tasks? More comprehensive evidence would make the generalization claim more robust.

5. **Do you have plans to expand the ablation study to cover all tasks?** The current ablation study only includes three out of the five tasks, limiting its scope in evaluating the framework’s modularity and adaptability. Would you consider conducting a more comprehensive ablation study that covers all tasks? This would provide a fuller picture of each module’s contributions and reveal potential strengths and limitations in greater detail.

6. **How would the Penetrative AI prompt perform if it incorporated only the enriched background information component of your framework?** The Penetrative AI paper emphasizes the importance of background information, and your proposal includes a component specifically designed to look up and integrate such background knowledge. How would the Penetrative AI prompt perform if it were enhanced with this enriched data alone, without additional components like data simplification or domain-specific adjustments? This comparison could help isolate the impact of enriched background information and clarify the unique contributions of each component in your framework.

7. **How do you justify the claim that these IoT tasks represent “physical laws”?** In the introduction and conclusion, the paper references “physical laws” as being relevant to IoT data, but this notion isn’t clearly supported in the experiments or throughout the paper. Could you clarify what is meant by “physical laws” in the context of these tasks? While IoT data inherently comes from physical processes, the specific connection to physical laws isn’t demonstrated or explained in the work. What role do physical laws play here, and how is this concept integrated into your framework?

---

### Note · Authors · 2024-11-13

I have read and agree with the venue's withdrawal policy on behalf of myself and my co-authors.